# Antibiotic Resistant Bloodstream Infections in Pediatric Patients Receiving Chemotherapy or Hematopoietic Stem Cell Transplant: Factors Associated with Development of Resistance, Intensive Care Admission and Mortality

**DOI:** 10.3390/antibiotics10030266

**Published:** 2021-03-05

**Authors:** Elio Castagnola, Francesca Bagnasco, Alessio Mesini, Philipp K. A. Agyeman, Roland A. Ammann, Fabianne Carlesse, Maria Elena Santolaya de Pablo, Andreas H. Groll, Gabrielle M. Haeusler, Thomas Lehrnbecher, Arne Simon, Maria Rosaria D’Amico, Austin Duong, Evgeny A. Idelevich, Marie Luckowitsch, Mariaclaudia Meli, Giuseppe Menna, Sasha Palmert, Giovanna Russo, Marco Sarno, Galina Solopova, Annalisa Tondo, Yona Traubici, Lillian Sung

**Affiliations:** 1Infectious Diseases Unit, IRCCS Istituto Giannina Gaslini, 16147 Genova, Italy; alessiomesini@gaslini.org; 2Epidemiology and Biostatistics Unit, IRCCS Istituto Giannina Gaslini, 16147 Genova, Italy; francescabagnasco@gaslini.org; 3Department of Pediatrics, Inselspital, Bern University Hospital, University of Bern, 3010 Bern, Switzerland; philipp.agyeman@insel.ch; 4Pediatric Hematology/Oncology, Department of Pediatrics, Inselspital, Bern University Hospital, University of Bern, 3010 Bern, Switzerland; roland.ammann@insel.ch; 5Kinderaerzte KurWerk, 3400 Burgdorf, Switzerland; 6Pediatric Oncology Institute, GRAACC/Federal University of Sao Paulo, Sao Paulo 04023-062, Brazil; fabiannecarlesse@graacc.org.br; 7Facultad de Medicina, Universidad de Chile, Hospital Dr. Luis Calvo Mackenna, Santiago 7500539, Chile; msantola@med.uchile.cl; 8Infectious Disease Research Program, Center for Bone Marrow Transplantation and Department of Paediatric Haematology and Oncology, University Children’s Hospital Muenster, 48149 Muenster, Germany; andreas.groll@ukmuenster.de; 9The Paediatric Integrated Cancer Service, Parkville, VIC 3052, Australia; gabriellehaeusler@petermac.org; 10Department of Infectious Diseases, Peter MacCallum Cancer Centre, Melbourne, VIC 3000, Australia; 11Sir Peter MacCallum Department of Oncology, University of Melbourne, Parkville, VIC 3010, Australia; 12Division of Pediatric Hematology and Oncology, Hospital for Children and Adolescents, University Hospital, Johann Wolfgang Goethe University, 60323 Frankfurt am Main, Germany; thomas.lehrnbecher@kgu.de (T.L.); marie.luckowitsch@kgu.de (M.L.); 13Pediatric Oncology and Hematology, Children’s Hospital Medical Center, University Clinics, 6642 Homburg, Germany; arne.simon@uks.eu; 14Department Hemato-Oncology, AORN Santobono-Pausilipon, 80129 Napoli, Italy; mrdamico@tiscali.it (M.R.D.); giumenna56@libero.it (G.M.); 15Child Health Evaluative Sciences, The Hospital for Sick Children, Peter Gilgan Centre for Research & Learning, Toronto, ON M5G 0A4, Canada; austinduong8@gmail.com (A.D.); palmerts@mcmaster.ca (S.P.); yona.traubici@rogers.com (Y.T.); lillian.sung@sickkids.ca (L.S.); 16Institute of Medical Microbiology, University Hospital Munster, 48149 Munster, Germany; evgeny.idelevich@med.uni-greifswald.de; 17Pediatric Hemato-Oncology Unit, Department of Clinical and Experimental Medicine, University of Catania, 95123 Catania, Italy; mclaudiameli@gmail.com (M.M.); diberuss@unict.it (G.R.); 18Pediatric Unit, Santa Maria Delle Grazie Hospital, ASL Napoli 2 Nord, Pozzuoli, 80027 Napoli, Italy; marco.sarno@aslnapoli2nord.it; 19Dmitry Rogachev Federal Scientific-Clinical Center of Children’s Hematology, Oncology and Immunology, 117997 Moscow, Russia; galina.solopova@fccho-moscow.ru; 20Paediatric Haematology/Oncology Department, Meyer Children’s University Hospital, 50134 Florence, Italy; annalisa.tondo@unifi.it

**Keywords:** antibiotic resistance, intensive care admission and mortality, bloodstream infections, pediatric patients, chemotherapy, allogeneic stem cell transplant

## Abstract

Bloodstream infections (BSI) are a severe complication of antineoplastic chemotherapy or hematopoietic stem cell transplantation (HSCT), especially in the presence of antibiotic resistance (AR). A multinational, multicenter retrospective study in patients aged ≤ 18 years, treated with chemotherapy or HSCT from 2015 to 2017 was implemented to analyze AR among non-common skin commensals BSI. Risk factors associated with AR, intensive care unit (ICU) admission and mortality were analyzed by multilevel mixed effects or standard logistic regressions. A total of 1291 BSIs with 1379 strains were reported in 1031 patients. Among Gram-negatives more than 20% were resistant to ceftazidime, cefepime, piperacillin-tazobactam and ciprofloxacin while 9% was resistant to meropenem. Methicillin-resistance was observed in 17% of *S. aureus* and vancomycin resistance in 40% of *E. faecium*. Previous exposure to antibiotics, especially to carbapenems, was significantly associated with resistant Gram-negative BSI while previous colonization with methicillin-resistant *S. aureus* was associated with BSI due to this pathogen. Hematological malignancies, neutropenia and Gram-negatives resistant to >3 antibiotics were significantly associated with higher risk of ICU admission. Underlying disease in relapse/progression, previous exposure to antibiotics, and need of ICU admission were significantly associated with mortality. Center-level variation showed a greater impact on AR, while patient-level variation had more effect on ICU admission and mortality. Previous exposure to antibiotics or colonization by resistant pathogens can be the cause of AR BSI. Resistant Gram-negatives are significantly associated with ICU admission and mortality, with a significant role for the treating center too. The significant evidence of center-level variations on AR, ICU admission and mortality, stress the need for careful local antibiotic stewardship and infection control programs.

## 1. Introduction

Infections represent important complications in pediatric patients receiving antineoplastic chemotherapy, or allogeneic hematopoietic stem cell transplantation (HSCT). The introduction of empirical antibacterial therapy with the combination of an anti-pseudomonal beta-lactam and an aminoglycoside in febrile neutropenic cancer patients has significantly decreased mortality [1,2]. Following the results of a recent a meta-analysis [3] monotherapy with beta-lactams active against Gram-negatives (including *P. aeruginosa*) is now recommended for the management of febrile neutropenia in pediatric patients with cancer [4] for institutions where resistance rates in Gram-negative isolates are low. However, this approach requires careful epidemiologic surveillance and continual re-evaluation of empiric antibiotic regimens in light of evolving institutional microbial resistance patterns [4,5]. Antibiotic resistance is a worldwide problem, although geographic and institution-level differences are observed (https://atlas-surveillance.com/#/heatmap/resistance) (access on 4 March 2021). This phenomenon also affects pediatric patients receiving antineoplastic chemotherapy or allogeneic HSCT [6] who become at risk of receiving an inadequate initial empirical therapy of febrile neutropenia, with an increased likelihood of complicated clinical course [7,8,9,10,11,12,13]. Knowledge of the epidemiology of antibiotic resistant bacterial infections and their consequences in pediatric cancer and HSCT patients is therefore mandatory in order to identify the best management strategies.

Aims of the present study were to describe the proportion of antibiotic resistant non-common skin contaminants causing bloodstream infections (BSI) in pediatric patients receiving antineoplastic treatments or HSCT, to describe clinical risk factors associated with development of antibiotic resistance and to the risk of complicated clinical course, i.e., intensive care unit (ICU) admission and death.

## 2. Materials and Methods

The study was a retrospective chart review conducted in centers located in Australia (*n* = 1), Brazil (*n* = 1), Canada (*n* = 1), Chile (*n* = 1), Germany (*n* = 3), Italy (*n* = 4), Russian Federation (*n* = 1) and Switzerland (*n* = 3). The research ethics board approval was obtained at each site where it was required, according to local regulations.

Inclusion criteria were a BSI diagnosed between 1 January 2015 and 31 December 2017 in patients aged ≤ 18 years, with a disease treated with chemotherapy or allogeneic HSCT, and due to Gram-positive (*S. aureus*, *E. faecalis* and *E. faecium*, viridans streptococci) or Gram-negative rods or yeasts. Episodes due to common skin contaminants (https://www.cdc.gov/nhsn/XLS/Common-Skin-Contaminant-List-June-2011.xlsx) (access on 4 March 2021) were excluded in order to avoid the bias of possible cases of contaminated blood cultures erroneously classified as infections and consequently to know the real impact of antibiotic resistance among “true pathogens” causing BSI.

For each BSI episode, patient-level demographic and disease-related variables were age (years), sex, type of underlying disease and the possible presence of relapse/progression, reception of autologous or allogeneic HSCT and post-transplant phases (pre-engraftment, presence of acute or chronic graft versus host disease). The type of underlying disease was classified into three major groups: (1) hematologic malignancy (HM) including acute lymphoblastic leukemia, acute myeloid leukemia, non-Hodgkin lymphoma, hemophagocytic lymphohistiocytosis and other leukemias; (2) solid tumors (ST) including neuroblastoma, bone or soft tissue sarcoma, central nervous system tumor, Hodgkin disease and other solid tumors; (3) non-malignant diseases (NMD) receiving allogeneic HSCT (bone marrow failure, primary immunodeficiency, and inborn errors of metabolism). Episode-level variables were administration of antibiotics (prophylaxis or therapy) in the 30 days preceding the episode, bacterial infection or colonization by the same pathogen causing the BSI in the three months preceding the episode, presence of neutropenia (absolute neutrophil count (ANC) < 500/μL) at diagnosis of the episode, administration of anti-pseudomonal empirical therapy for febrile neutropenia (piperacillin/tazobactam, ceftazidime or cefepime, monotherapy or combined with an aminoglycoside) at the time of BSI, ICU admission for BSI, and death within 30 days from the episode.

For each isolated strain’s antibiotic, susceptibility to the following drugs was registered: *S. aureus*: methicillin, vancomycin, daptomycin, linezolid, tigecycline, ceftaroline or ceftobiprole; *E. faecalis* and *E. faecium*: ampicillin, vancomycin, teicoplanin, daptomycin, linezolid, tigecycline; viridans streptococci: ampicillin or penicillin; Gram-negatives: meropenem or other carbapenems, colistin, amikacin, gentamycin, tobramycin, ciprofloxacin, ceftazidime, cefepime, piperacillin-tazobactam, tigecycline, ceftolozane-tazobactam and ceftazidime-avibactam; *Candida* spp.: fluconazole, caspofungin and micafungin. Pathogens were recorded as susceptible or resistant according to the local microbiology laboratory classifications following EUCAST or CLSI methodologies and criteria [14,15], since the minimum inhibitory concentrations were not consistently available. In the case of the definition of intermediate or dose-dependent susceptibility, the strain was recorded as susceptible.

Data were collected at each center by trained personnel and registered in a secure web-based database using the Research Electronic Data Capture (REDCap) platform (www.project-redcap.org) (access on 4 March 2021) [16].

### Statistical Analysis

Categorical variables were reported as absolute frequencies and percentages. Continuous data were reported as the median and interquartile range (IQR), due to their non-normal (Gaussian) distribution. Percentages of antibiotic-resistant infections by pathogen were calculated with a 95% confidence interval (CI) and reported with the robust estimator of variance allowing for intra-group correlation due to centers. The association between binary outcome variables (antibiotic resistant BSI, ICU admission or death) and independent variables was assessed by multilevel (three or two levels) mixed effects logistic regressions [17], or by standard logistic regression and reported in terms of the odds ratio (OR) and 95% CI. The three-level model had two random-effects equations, the first was a random intercept at the center level, and the second was a random intercept at the patient level (nested in the center level). Multivariable regressions for the likelihood of antibiotic resistance were focused on groups or single pathogens more representative in terms of the tested susceptibility and frequency of antibiotic resistance. The demographic and clinical characteristics of patients at the time of the BSI were entered into the multivariable models. For the likelihood of ICU admission and death, the antibiotic resistance of the Gram-negatives was also included considering their resistance to 1, 2–3 and 4–5 drugs among meropenem, amikacin, ciprofloxacin, ceftazidime, and piperacillin-tazobactam. A likelihood-ratio test (LR) was used to measure the effect of each predictor and to compare multilevel mixed effects logistic model versus standard logistic regression that was performed in case of a statistically insignificant LR test. All tests were two-tailed and a *p* value < 0.05 was considered statistically significant. All analyses were performed using Stata (StataCorp. Stata Statistical Software, Release 13.1, College Station, TX, USA, StataCorporation, 2013).

## 3. Results

A total of 1340 BSIs were registered, but 49 (3.6%) were not eligible since they occurred in patients > 18 years old (*n* = 18), or after December 31, 2017 (*n* = 24); age > 18 and year > 2017 (*n* = 1) or pathogen were not recorded (*n* = 6). The analysis was therefore performed on 1291 BSIs observed in 1031 patients, 840 (81.5%) with a single episode, 149 (14.4%) with two episodes and 42 (4.1%) with ≥ three episodes. BSIs were observed in 756 (58.6%) males and the median age at BSI was eight (IQR 3–13) years. The most frequent underlying condition was HM (*n* = 838, 64.9%), mainly acute lymphoblastic leukemia (*n* = 457, 54.5%), while in 320 (24.8%) BSIs followed HSCT, mainly allogeneic (83.1%). Appendix A reports demographic and clinical characteristics at the time of BSI. Overall, 1210 (93.7%) episodes were single-agent (723 Gram-negatives, 402 Gram-positives and 85 fungi) and 81 (6.3%) polymicrobial (74 two-agent and 7 three-agents) for a total of 1379 strains. Among the strains recorded 1289 (93.5%) were bacteria, 831 (64.5%) Gram-negatives, 458 (35.5%) Gram-positives, and 90 (6.5%) fungi. A complete list of the isolated pathogens is available in Appendix A. *E. coli* (20.5%) was the most frequently isolated pathogen, followed by *S. aureus* (13.5%), *K. pneumoniae* (12.8%), viridans streptococci (12.0%) and *P. aeruginosa* (10.2%). *A. baumannii* complex (2.2%), *S. maltophilia* (2.5%), *B. cepacia* (0.5%), and anaerobes (0.1%) were rare. *Candida* (90.0%) was the most frequently isolated yeast genus and *C. parapsilosis* (30.0%) was the most frequently isolates species. No case of *C. auris* was registered.

### 3.1. Resistance to Anti-Infectives

Table 1 summarizes the proportions of antibiotic-resistant Gram-negatives and Gram-positives, while data for specific pathogens and antibiotics are shown in Figure 1 and Figure 2 and detailed in Appendix A. Among Gram-negatives, the proportion of resistant strains was similar for ceftazidime, cefepime, and ciprofloxacin (29.5%, 25.8% and 25.5%, respectively) and a little lower for piperacillin-tazobactam (21.8%). The overall amikacin resistance was 7.5% but was higher for *P. aeruginosa* (15.4%), *A. baumannii* complex (30.4%) and *K. pneumoniae* (14.6%). Globally, meropenem resistance was 9.0%, with higher proportions for *P. aeruginosa* (27.3%), *A. baumannii* complex (25.0%), and *K. pneumoniae* (15.9%). Among the 489 *Enterobacteriales* tested, resistance was 5.9% (*n* = 29) (Appendix A). As for Gram-positives, methicillin resistance was 16.8% for *S. aureus* (MRSA) without any resistance to vancomycin. Among enterococci, vancomycin resistance was 26.8%, but it was highest in *E. faecium* (39.5%). Among *Candida* strains, fluconazole resistance was 27.0% (95% CI 16.2–41.5). Data on yeast resistance to antifungal are detailed in Appendix A.

Table 2 and Table 3 report on risk factors for Gram-negative and Gram-positive antibiotic resistant BSIs, respectively. Previous exposure to antibiotics or previous infectious episode due to the same Gram-negative were significantly associated with a higher risk of resistance to all antibiotics except gentamycin (Table 2). Previous exposure to carbapenems was significant for resistance for many antibiotics, and especially to meropenem. No effect was observed for previous colonization on the development of a resistant Gram-negative BSI, except for piperacillin-tazobactam. A center-level variation was observable for all antibiotics, while variation for patients nested within the center was observable for meropenem, gentamycin, ceftazidime, and piperacillin-tazobactam (Table 2). Conversely, among Gram-positives (Table 3), previous colonization was significantly associated with a higher risk of MRSA BSI. A significant effect of the treating center was observable for MRSA or ampicillin-resistant enterococci (Table 3). Appendix A show proportions of antibiotic-resistant strains stratified by treating center.

### 3.2. Admission in Intensive Care Unit and Mortality

Table 4 reports risk factors for ICU admission or death. A total of 171 (13.2%) episodes required ICU admission. HM had a greater risk of ICU admission as well as BSI in the presence of neutropenia, previous exposure to carbapenems and BSI due to Gram-negatives resistant to >3 antibiotics. Overall, death was reported for 99 (7.7%) episodes and in 67 (67.7%) it was attributed to BSI. The underlying disease in relapse/progression, a previous exposure to antibiotics (mainly carbapenems or combination therapy), and a need for ICU admission for BSI were significantly associated with mortality. Patient-level variation showed a greater impact on ICU admission and mortality than center-level variation.

## 4. Discussion

In this multicenter, multinational retrospective study, we collected 1291 BSIs due to non-common skin contaminants occurring in pediatric patients treated with chemotherapy or allogeneic HSCT to study proportions of resistant strains and risk factors for antibiotic resistance, ICU admission, or death.

Resistance to antibiotics was high among Gram-negatives being approximately 25% for ceftazidime and cefepime and near 20% for piperacillin-tazobactam. Resistance to meropenem was < 10%, but was higher for *K. pneumoniae* (15.9%), *P. aeruginosa* (27.3%), and *A. baumannii* complex (25.0%), the 3rd, 5th, and 9th most frequently reported pathogens, respectively. These proportions of resistant strains are worrisome since they regard the antibiotics generally recommended as monotherapy for empirical treatment of febrile neutropenia in pediatric patients [2,4], with the consequent non-negligible risk of treatment failure. Previous exposure to antibiotics, with a highest risk for carbapenems [18,19], was significantly associated with antibiotic-resistant Gram-negative BSI, as reported in adults [20]. Finally, a BSI due to Gram-negatives resistant to >3 antibiotics was significantly associated with ICU admission and death. It is noteworthy that a previous colonization or infection by the same pathogen did not affect ICU admission or death, perhaps since the choice of empirical therapy in case of febrile neutropenia in a colonized patient could have been guided by this information. The study also showed that about 25% of Gram-negatives were resistant to ciprofloxacin. This finding indicates the need for a rethink on the fluoroquinolone prophylaxis of febrile neutropenia, which, despite some effectiveness during chemotherapy courses (but not pre-engraftment neutropenia), has been associated with increased antibiotic resistance [21,22,23,24]. In this regard, it should be noted that in the most recent pediatric guidelines this procedure has received a weak recommendation about its use [25]. Multilevel mixed-effects logistic regressions showed an evident center-level variation on antibiotic resistance among Gram-negatives. Taken together, all of these observations emphasize the need for the establishment at the local level of antimicrobial stewardship and infection prevention and control programs [4,5,25,26,27], which could also have a favorable impact on the management of infectious episodes and their complications. *S.aureus* was the second most frequently reported pathogen, with MRSA detected in near 1/6 of cases; VRE represented near 40% of *E.faecium* strains, but was about 4% in *E.faecalis*, while penicillin/ampicillin resistance was frequent among viridans streptococci (28 and 41.4%, respectively). MRSA colonization was a significant risk factor for resistant BSI, similarly to what is generally observed for surgical site infections [28]. Finally, yeasts represented an infrequent cause of BSI in this patient population.

Conditions related to the underlying disease and its treatment also had a significant impact on ICU admission and mortality: HM was significantly associated with the risk of ICU admission, as well as a BSI developing during neutropenia alongside a relapsing/resistant disease or ICU admission was associated with an increased risk of death.

This study represents the largest available series on antimicrobial susceptibility of non-common skin-contaminant bacteria causing BSI in pediatric patients receiving chemotherapy or HSCT collected from different parts of the world and provides important information on the burden of antibiotic resistance in this patient population and its relationship with complicated clinical course, but it also has important limitations. The choice of not collecting data on BSI due to common skin contaminants permitted a better understanding of the phenomenon of antibiotic resistance and its consequences, but could have biased the results at least partially, especially after observations of infections due to multidrug resistant coagulase-negative staphylococci [29]. Moreover, we do not know if the carbapenem resistance we observed was due to carbapenemases (and if so, which ones) or other mechanisms. Knowledge of this aspect could have important implications since new antibiotics as ceftazidime-avibactam or meropenem-vaborbactam [30,31] are not effective against some carbapenemases frequently identified in pediatric patients [26], while ceftolozane-tazobactam could have some effectiveness against these strains [32]. Unfortunately, while some pediatric pharmacological data are available for ceftazidime-avibactam [33] and ceftolozane-tazobactam [34,35], they are scarce and fragmented, when not available at all, for meropenem-vaborbactam [36], cefiderocol [37], and cefepime-zidebactam [38], drugs that could be effective against bacteria resistant to the other antibiotics. We do not have data on the in vitro effectiveness of the new antibiotics in our patient population since they were tested in a negligible proportion of strains, if any, maybe because of scarce availability and/or the restriction or absence of authorizations in pediatrics. This is another limitation. Due to the multinational nature of the study, we were reliant on local antimicrobial cultures, susceptibility testing techniques, and reporting. While this variation could impact our results, the large number of episodes involved maintains the generalizability of the results.

Finally, this study showed the significant effect of local conditions on the development of antibiotic resistant BSI and unfavorable outcomes. Local antimicrobial stewardship and adherence to infection control programs are mandatory in order to reduce the spread of resistant pathogens and the unnecessary use of antibiotics, as well as studies on new antibiotics in order to quickly offer the best therapeutic strategies in children with underlying diseases that could expose the patient to severe infections and their complications.

## Figures and Tables

**Figure 1 antibiotics-10-00266-f001:**
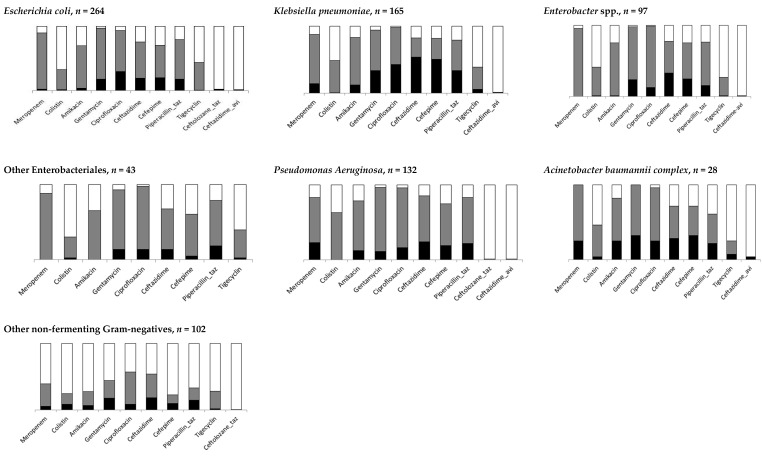
Distribution of resistant antibiotic bacteremia by pathogen type (details in Appendix A) of Gram-negative isolates. Black box represents percentages of resistant bacteremia, gray box represents susceptible bacteremia, and white box for bacteremia not tested.

**Figure 2 antibiotics-10-00266-f002:**
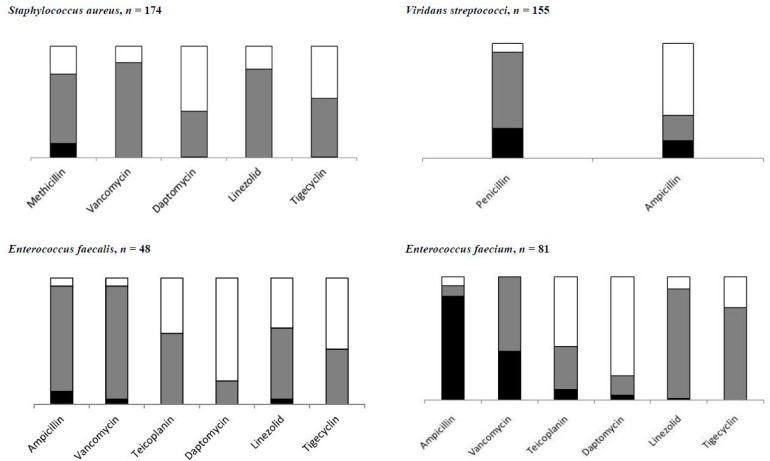
Distribution of resistant antibiotic infections by pathogen type (details in Appendix A) of Gram-positive. Black box represents percentages of resistant antibiotic infections, gray box of susceptible antibiotic infections, and white box for antibiotic infections not tested.

**Table 1 antibiotics-10-00266-t001:** Proportions of strains resistant to the antibiotics most frequently used against Gram-negatives and for specific Gram-positives.

	Resistant, *n*	% Resistance (95% CI)
**Gram-negatives, *n* = 797**		
Meropenem	72	9.0 (3.7–20.5)
Amikacin	60	7.5 (3.1–17.0)
Gentamycin	173	21.7 (11.8–36.5)
Ceftazidime	235	29.5 (14.2–51.4)
Cefepime	206	25.8 (10.0–52.4)
Piperacillin-tazobactam	174	21.8 (16.8–27.8)
Ciprofloxacin	203	25.5 (14.2–41.4)
***S.aureus, n* = 131**		
Methicillin	22	16.8 (7.9–32.1)
**Enterococci, *n* = 127**		
Ampicillin	73	57.5 (28.0–82.4)
Vancomycin	34	26.8 (13.4–46.4)
**Viridans streptococci**		
Penicillin, ***n* = 143**	40	28.0 (18.7–39.6)
Ampicillin, ***n* = 58**	24	41.4 (28.1–56.1)

**Table 2 antibiotics-10-00266-t002:** Multivariable logistic regression models for BSI due to antibiotic resistant Gram-negatives (*n* = 797).

	Odds Ratio (95% Confidence Interval)						
Factors	Meropenem *	Amikacin **	Gentamycin *	Ciprofloxacin **	Ceftazidime *	Cefepime **	Piperacillin-Tazobactam *
**Sex**, *p*-value	0.768	0.981	0.327	0.204	0.316	0.296	0.013
Male vs. female	0.9 (0.4–1.9)	1.0 (0.5–1.8)	1.3 (0.7–2.3)	1.3 (0.9–1.9)	0.7 (0.4–1.3)	1.3 (0.8–2.0)	0.6 (0.3–0.9)
**Age at bloodstream infections**, years, *p*-value	0.9 (0.8–1.1), 0.062	1.0 (0.9–1.1), 0.383	1.0 (0.9–1.1), 0.344	1.0 (1.0–1.1), 0.017	1.0 (0.9–1.1), 0.324	1.0 (0.9–1.1), 0.120	1.0 (0.9–1.1), 0.654
**Underlying disease**, *p*-value	0.038	0.015	0.706	0.061	0.195	0.438	0.476
NMD vs. HM	4.0 (1.1–14.0)	3.5 (1.5–8.0)	1.5 (0.6–3.7)	2.0 (1.0–3.7)	1.4 (0.5–3.5)	1.2 (0.6–2.4)	1.6 (0.7–3.5)
ST vs. HM	0.8 (0.3–2.4)	1.0 (0.4–2.4)	1.0 (0.5–2.1)	0.8 (0.5–1.4)	0.6 (0.3–1.2)	0.7 (0.4–1.3)	1.2 (0.7–2.2)
**Allogeneic stem cell transplant**, *p*-value	0.533	0.611	0.199	0.773	0.927	0.443	0.382
Yes vs. no	1.3 (0.5–3.3)	0.8 (0.4–1.8)	1.8 (0.9–3.7)	0.9 (0.6–1.6)	1.0 (0.5–2.1)	0.8 (0.4–1.4)	1.3 (0.7–2.4)
**Relapse/ progression**, *p*-value	0.279	0.438	0.150	0.099	0.218	0.008	0.467
Yes vs. no	1.6 (0.7–3.9)	1.3 (0.7–2.6)	1.6 (0.8–3.0)	1.4 (0.9–2.2)	1.5 (0.8–2.9)	2.0 (1.2–3.3)	1.2 (0.7–2.1)
**BSI**, *p*-value	0.768	0.298	0.668	0.392	0.216	0.280	0.028
Single agent vs. polymicrobial	0.8 (0.1–4.0)	0.5 (0.1–1.8)	0.8 (0.3–2.1)	1.4 (0.6–3.4)	0.5 (0.2–1.4)	0.6 (0.2–1.5)	0.4 (0.2–0.9)
**Previous antibacterial exposure (prophylaxis/therapy)**^1^, *p*-value	<0.001	<0.001	0.138	0.008	0.009	0.215	<0.001
Standard regimen vs. none	5.1 (1.5–17.4)	4.5 (1.8–11.4)	2.1 (1.1–4.2)	1.7 (1.0–2.8)	2.2 (1.1–4.6)	1.7 (0.9–2.9)	3.3 (1.7–6.3)
Carbapenems vs. none	31.5 (5.1–193.4)	7.3 (2.6–20.1)	2.0 (0.8–4.7)	2.5 (1.4–4.7)	3.8 (1.4–10.2)	1.9 (0.9–3.8)	3.4 (1.5–7.4)
Fluoroquinolones/β-lactams/Combination ^2^/Others vs. none	5.8 (1.3–25.7)	2.2 (0.7–6.9)	1.4 (0.6–3.3)	2.1 (1.1–3.7)	1.4 (0.6–3.5)	1.4 (0.7–2.9)	2.3 (1.1–4.9)
**Neutropenia**, *p*-value	0.016	0.494	0.485	0.427	0.872	0.190	0.048
Yes vs. no	3.1 (1.1–8.9)	1.3 (0.6–2.7)	1.3 (0.7–2.4)	1.2 (0.8–1.9)	0.9 (0.5–1.8)	1.4 (0.8–2.4)	1.7 (1.0–3.0)
**Previous colonization**, *p*-value	0.120	0.257	0.895	0.891	0.421	0.653	0.035
No vs. yes	2.2 (0.8–5.7)	1.6 (0.7–3.5)	1.0 (0.5–2.2)	1.0 (0.6–1.8)	1.4 (0.6–2.9)	1.1 (0.6–2.0)	2.0 (1.0–3.8)
**Previous infection**, *p*-value	0.161	0.040	0.280	<0.001	<0.001	<0.001	0.031
No vs. yes	1.9 (0.8–5.0)	2.2 (1.1–4.6)	1.5 (0.7–3.3)	2.6 (1.5–4.4)	3.9 (1.7–8.9)	3.6 (1.8–6.9)	2.0 (1.1–3.7)
**Random effect, variance component, centre**	2.0 (0.4–9.1)	1.6 (0.4–6.2)	1.1 (0.3–3.7)	1.8 (0.7–4.7)	4.4 (1.3–14.6)	8.4 (2.9–24.9)	0.7 (0.2–2.6)
**Random effect, variance component, patient**	1.8 (0.1–28.9)	NA	2.9 (0.7–11.0)	NA	2.7 (0.6–12.5)	NA	1.3 (0.2–7.0)
**LR test vs. logistic regression**, *p*-value ***	<0.0001	<0.0001	<0.0001	<0.0001	<0.0001	<0.0001	0.0001

* Three-level mixed effects logistic regression with random effects for patients nested within centers. ** Two-level mixed effects logistic regression with random effects for centers. *** If *p*-value of likelihood-ratio test (LR) test comparing multilevel mixed effects logistic model versus standard logistic regression was not statistically significant, standard logistic regression was adopted. ^1^ Due to low numbers, Fluoroquinolones, β-lactams not active vs. *P. aeruginosa*, combination and other previous exposure were grouped together. ^2^ Combination of two or more of the following fluoroquinolone/β-lactams not active vs. *P. aeruginosa*/Standard regimen active vs. *P. aeruginosa*/carbapenem. NMD: Non-malignant disease receiving allogeneic stem cell transplant; HM: hematologic malignancy; ST: solid tumors; NA: not applicable.

**Table 3 antibiotics-10-00266-t003:** Multivariable logistic regression models for models for BSI due to antibiotic resistant among Gram-positives.

	Odds Ratio (95% Confidence Interval)				
Factors	Methicillin-*Staphylococcus aureus, n* = 131 ***	Penicillin-Viridians Streptococci, *n* = 143 **	Ampicillin-Viridians Streptococci, *n* = 58 **	Ampicillin-*Enterococcus* (*faecalis & faecium*), *n* = 127 *	Vancomycin-*Enterococcus*(*faecalis & faecium*), *n* = 127 **
**Sex**, *p*-value	0.760	0.792	0.228	0.490	0.197
Male vs. female	1.2 (0.3–4.9)	0.9 (0.4–2.0)	0.4 (0.1–1.7)	1.5 (0.5–4.5)	1.9 (0.7–5.1)
**Age at bloodstream infections,** years, *p*-value	1.0 (0.9–1.1), 0.868	0.9 (0.8–1.1), 0.107	0.9 (0.8–1.1), 0.118	1.0 (0.9–1.1), 0.790	1.0 (0.9–1.1), 0.790
**Underlying disease**, *p*-value	0.462	0.418	0.947	0.072	0.570
NMD vs. HM	1.4 (0.2–9.7)	1.0 (0.1–9.8)	NA	0.1 (0.0–1.2)	0.4 (0.1–4.9)
ST vs. HM	2.7 (0.5–13.6)	0.4 (0.1–1.8)	0.9 (0.1–5.8)	0.4 (0.1–1.6)	0.7 (0.2–2.9)
**Allogeneic stem cell transplant**, *p*-value	0.324	0.693	0.461	0.046	0.796
Yes vs. no	2.9 (0.3–24.4)	1.3 (0.4–4.6)	2.2 (0.5–19.5)	5.2 (0.9–29.2)	1.2 (0.3–4.4)
**Relapse/ progression**, *p*-value	0.278	0.890	0.498	0.282	0.550
Yes vs. no	0.3 (0.1–2.4)	0.9 (0.3–2.7)	0.5 (0.1–3.1)	2.0 (0.6–7.4)	1.3 (0.5–3.6)
**BSI**, *p*-value	0.176	0.447	0.879	0.895	0.925
Single agent vs. polymicrobial	0.1 (0.0–2.4)	0.6 (0.2–2.0)	1.2 (0.2–8.5)	1.1 (0.2–6.1)	0.9 (0.2–4.8)
**Previous antibacterial exposure (prophylaxis/therapy)**, *p*-value			0.004		
Yes vs. no	NA	NA	7.7 (1.7–35.5)	NA	NA
**Previous antibacterial exposure (prophylaxis/therapy) ^1^**, *p*-value	0.068	0.489		0.396	0.083
Standard regimen vs. no one	6.4 (1.1–39.5)	1.8 (0.6–5.6)	NA	1.1 (0.2–7.1)	3.2 (0.4–25.5)
Carbapenem vs. no one	NA	1.6 (0.4–6.1)	NA	1.3 (0.2–8.2)	8.7 (0.8–90.7)
Fluoroquinolones/β-lactams/Combination ^2^/Others vs. no one	4.5 (0.8–26.7)	2.2 (0.7–6.7)	NA	3.6 (0.5–24.5)	2.5 (0.2–25.2)
**Neutropenia**, *p*-value	0.656	0.840	0.617	0.047	0.048
Yes vs. no	0.7 (0.2–3.1)	1.1 (0.3–4.4)	0.6 (0.1–4.1)	3.7 (0.9–14.5)	3.5 (1.1–11.1)
**Previous colonization**, *p*-value	0.013	0.414	0.918	0.346	0.073
Yes vs. no	6.7 (1.4–31.3)	0.3 (0.1–4.6)	0.9 (0.1–10.8)	2.2 (0.4–11.9)	2.6 (0.9–7.4)
**Previous infection**, *p*-value		0.556	0.725	0.021	0.726
Yes vs. no	NA	1.5 (0.4–6.4)	0.6 (0.1–7.9)	5.7 (1.1–28.8)	0.8 (0.2–2.7)
**Random effect, variance component, centre**	2.7 (0.3–22.7)	NA	NA	1.5 (0.3–7.0)	NA
**LR test vs. logistic regression**, *p*-value *******	0.0255	0.5326	1.000	0.0002	0.6495

* Two-level mixed effects logistic regression with random effects for centers. ** Standard logistic regression *** If *p*-value of likelihood-ratio test (LR) test comparing multilevel mixed effects logistic model versus standard logistic regression was not statistically significant, standard logistic regression was performed. ^1^ Due to low numbers, Fluoroquinolones, β-lactams not active vs. *P. aeruginosa*, combination and other previous exposure were grouped into one group. ^2^ Combination of two or more of the following fluoroquinolone/β-lactams not active vs. *P. aeruginosa*/Standard regimen active vs. *P. aeruginosa*/carbapenem. NMD: Non-malignant disease receiving allogeneic stem cell transplant; HM: hematologic malignancy; ST: solid tumors; NA: not applicable.

**Table 4 antibiotics-10-00266-t004:** Multivariable logistic regression models for ICU admission or mortality during BSI.

Factors	Odds Ratio (95% Confidence Interval)	
	ICU Admission *	Mortality *
**Sex**, *p*-value	0.302	0.724
Male vs. female	0.7 (0.4–1.3)	0.9 (0.4–1.9)
**Age at bloodstream infections**, years, *p*-value	1.0 (0.9–1.1), 0.210	0.9 (0.8–1.1), 0.068
**Underlying disease**, *p*-value	0.018	0.098
NMD vs. HM	0.8 (0.3–2.2)	3.6 (1.0–13.4)
ST vs. HM	0.3 (0.1–0.8)	1.2 (0.4–3.3)
**Relapse/ progression**, *p*-valueYes vs. no	0.2701.5 (0.7–2.9)	0.0045.3 (1.7–16.5)
**Allogeneic stem cell transplant phase**, *p*-value	0.5778	0.089
Pre-engraftment vs. no allogenic-HSCT	1.3 (0.5–3.0)	0.8 (0.2–2.9)
Acute GvHD vs. no allogenic-HSCT	2.1 (0.5–8.9)	1.7 (0.3–10.8)
Chronic GvHD vs. no allogenic-HSCT	1.8 (0.3–11.3)	7.0 (0.9–51.6)
Post-engraftment vs. no allogenic-HSCT	0.6 (0.1–2.1)	4.2 (1.0–17.8)
**Neutropenia**, *p*-value	0.023	0.327
Yes vs. no	2.5 (1.2–5.3)	1.6 (0.6–4.1)
**Previous antibacterial exposure (prophylaxis/therapy)**^1^, *p*-value	0.267	0.002
Fluoroquinolones vs. no one/β-lactams	1.4 (0.3–6.8)	2.5 (0.3–21.0)
Standard regimen vs. no one/β-lactams	1.4 (0.7–2.8)	0.9 (0.3–2.7)
Carbapenem vs. no one/β-lactams	2.8 (1.2–6.7)	3.8 (1.1–13.5)
Combination ^2^ vs. no one/β-lactams	1.1 (0.1–8.4)	9.1 (1.1–77.8)
Others vs. no one/β-lactams	2.0 (0.6–6.6)	8.2 (1.6–41.9)
Previous colonization, *p*-value	0.193	0.265
Yes vs. no	0.6 (0.2–1.3)	1.8 (0.6–5.3)
**Previous infection**, *p*-value	0.835	0.095
Yes vs. no	0.9 (0.4–2.0)	2.7 (0.8–8.9)
BSI, *p*-value	0.936	0.441
Single agent vs. polymicrobial	0.9 (0.3–2.8)	0.5 (0.1–2.6)
**Gram-negatives antibiotic resistance**, *p*-value	<0.001	0.167
Gram-negatives resistant to 1 antibiotic ^3^ vs. susceptible	0.3 (0.1–0.8)	3.4 (0.8–13.9)
Gram-negatives resistant to 2 or 3 antibiotics ^3^ vs. susceptible	0.7 (0.3–1.9)	3.7 (1.0–13.7)
Gram-negatives resistant to 4 or 5 antibiotics ^3^ vs. susceptible	18.0 (3.7–87.2)	4.5 (1.0–20.0)
Not applicable vs. susceptible	0.8 (0.4–1.6)	2.3 (0.8–6.5)
**ICU for bloodstream infection**, *p*-value Yes vs. no	NA	<0.001,44.4 (7.6–258.5)
**Random effect, variance component, center**	1.8 (0.5–6.6)	0.8 (0.2–5.9)
**Random effect, variance component, patient**	4.4 (0.9–20.2)	3.8 (0.7–19.7)
**LR test vs. logistic regression**, *p*-value **	<0.0001	0.0172

* Three-level mixed effects logistic regression with random effects for patients nested within centers. ** If *p*-value of likelihood-ratio test (LR) test, comparing multilevel mixed effects logistic model versus standard logistic regression, was not statistically significant, standard logistic regression was performed. ^1^ β-lactams not active vs. *P. aeruginosa* was considered as reference group due to no observed events in this group. ^2^ Combination of two or more of the following fluoroquinolone/β-lactams not active vs. *P. aeruginosa*/Standard regimen active vs. *P. aeruginosa*/carbapenem. ^3^ Meropenem, amikacin, ciprofloxacin, ceftazidime and piperacillin-tazobactam. NMD: Non-malignant disease receiving allogeneic stem cell transplant; HM: Hematologic malignancy; ST: Solid tumors; NA: Not applicable.

## Data Availability

Data is contained within the article or Appendix A. They are stored in a REDCap database and can be available on request from the corresponding author. The data are not publicly available due to personal data protection rules.

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
