# Peer review of "Antibiotic Resistant Bloodstream Infections in Pediatric Patients Receiving Chemotherapy or Hematopoietic Stem Cell Transplant: Factors Associated with Development of Resistance, Intensive Care Admission and Mortality"

_antibiotics, 2021, doi:10.3390/antibiotics10030266_

Round 1

Reviewer 1 Report

This is a multicenter, multinational retrospective study that explores factors associated with antibiotic resistant bloodstream infections in pediatric cancer or hematopoetic stem cell transplant patients. The manuscript specifically shows that patients with hematologic malignancies, neutropenia, previous exposure to carbapenems, or bloodstream infections due to gram negative organisms resistant to greater than 3 antibiotics who develop bloodstream infections have a greater risk of ICU admissions. It also shows that bloodstream infection patients with cancer relapse/progression, previous exposure to antibiotics, and need for ICU admission are associated with mortality. The manuscript is a helpful addition to the literature that helps ascertain prognosis for patients who develop a bloodstream infection during cancer treatment. The manuscript also illustrates a considerable degree of resistance to antibiotics which are typically administered as monotherapy for empiric treatment of febrile neutropenia, which could shape future management of these circumstances.

Major points:

There are many grammatical errors throughout the entire manuscript. I have tried to point out many of these in my comments, but this manuscript needs to be edited by an English-language scientific editor prior to resubmission. There errors are most pronounced in the abstract, introduction, and discussion sections. There are also several typographic errors throughout the manuscript. The text needs to be carefully reviewed before resubmission. Many of the sentences in the introduction and discussion are difficult to understand because of excessive complex sentence structure and the use of multiple commas in single sentences creating run-ons. This is particularly notable in the discussion section where run-on sentences and sentences that start with “And” are noted.

Minor points:

Abstract: “Bloodstream infections (BSI) are severe complication of antineoplastic chemotherapy or stem cell transplant (SCT)” should say “Bloodstream infections (BSI) are severe complications of antineoplastic chemotherapy or stem cell transplant (SCT), especially in the presence of antibiotic resistance”

Abstract: “Among Gram-neg-atives more than 20% were resistant to ceftazidime, cefepime, piperacillin-tazobactam and ciprofloxacin while 9% was resistant to meropenem.” Should say “Among Gram-neg-atives more than 20% were resistant to ceftazidime, cefepime, piperacillin-tazobactam and ciprofloxacin while 9% were resistant to meropenem.

Abstract: The last part of the following sentence is not clear: “Previous exposure to antibiotics, especially to carbapenems, was significantly associated with resistant gram-negative BSI while previous colonization with methicillin-resistant S. aureus BSI.” Previous colonization with what was associated with MRSA BSI?

Abstract: “Previous exposure to antibiotics or colonization can be cause of antibiotic resistant BSI.” This sentence does not make sense and needs to be rewritten for clarity. Do you mean “Previous exposure to anitibiotics or bacterial colonization can be the cause of antibiotic resistant BSI”?

Abstract: “This aspect stresses the need for careful local antibiotic stewardship and infec-tion control programs.” What aspect?

Introduction: “Historical studies showed that empirical antibacterial therapy…” In what context? In patients receiving chemotherapy or just SCT patients? It is not clear the way it is written in the introduction.

Introduction: “Following the results of a recent a meta-analysis 3monotherapy” The 3rd reference is not appropriately bracketed in the text. It should be denoted as [3].

Introduction: “Antibiotic resistance (AR) is a worldwide threat, although vary considerably” It should say “varies” instead of “vary”

Introduction: “Antibiotic resistance (AR) is a worldwide threat, although vary considerably between institutions and geographical regions (https://atlas-surveillance.com/#/heatmap/resistance) and involves also . pediatric pa-tients receiving chemotherapy or allogeneic SCT, with consequent risk of inadequate ini-tial empirical therapy of febrile neutropenia [6] causing increased morbidity and mortality [7-13]. “This sentence is difficult to follow and should be rewritten for clarity.

Introduction: “Epidemiological knowledges on antibiotic resistant bacteria and the consequences of invasive infections due to these pathogens in pediatric cancer and SCT patients are therefore mandatory in order to decide the best management strategy.” This sentence needs to be rewritten using proper grammar – epidemiological knowledge of antibiotic resistant bacteria…

Introduction: “Aims of the present study were to describe the proportion of antibiotic resistant non-common skin contaminants causing bloodstream infections (BSI) in pediatric patients re-ceiving antineoplastic treatments or SCT, to describe clinical risk factors associated with development of antibiotic resistance and to the risk of complicated clinical course, i.e. in-tensive care unit (ICU) admission and mortality.” Please change to “Aims of the present study were to describe the proportion of antibiotic resistant non-common skin contaminants causing bloodstream infections (BSI) in pediatric patients re-ceiving antineoplastic treatments or SCT, to describe clinical risk factors associated with development of antibiotic resistance and the risk of a complicated clinical course, i.e. in-tensive care unit (ICU) admission and mortality.”

Materials and Methods: “For each BSI episode, demographic and disease-related variablese” There is a typo here with an “e” on the end of variables.

Discussion: “Resistance among Gram-negatives was also about 25% for ciprofloxacin, overcom-ing the need for rethinking the practice of fluoroquinolone prophylaxis of febrile neutro-penia [21-24], which has indeed weakly recommended in the most recent pediatric guide-lines[25].” Please rewrite this sentence for clarity. Do you think the prophylaxis is the reason for resistance in this population? Should the prophylaxis be administered or not? It is unclear based on the way this sentence is written.

Discussion: “As a whole, all these observations underlying the need of effective, local antimicrobial stewardship and infection prevention and control programs [4,5,25-27] that could have an effective impact in the management of an infectious episode, including complicated clinical course.” This sentence has a lot of commas and is fiddicult to understand. Please rewrite for clarity.

Author Response

Major points:

Thanks to the reviewer for the observation and the modifications suggested. Changes are written in red in the text

Minor points:

Questions

Answers

Abstract: “Bloodstream infections (BSI) are severe complication of antineoplastic chemotherapy or stem cell transplant (SCT)” should say “Bloodstream infections (BSI) are severe complications of antineoplastic chemotherapy or stem cell transplant (SCT), especially in the presence of antibiotic resistance”

The phrase has been modified

Abstract: “Among Gram-neg-atives more than 20% were resistant to ceftazidime, cefepime, piperacillin-tazobactam and ciprofloxacin while 9% was resistant to meropenem.” Should say “Among Gram-neg-atives more than 20% were resistant to ceftazidime, cefepime, piperacillin-tazobactam and ciprofloxacin while 9% were resistant to meropenem.

The phrase has been modified

Abstract: The last part of the following sentence is not clear: “Previous exposure to antibiotics, especially to carbapenems, was significantly associated with resistant gram-negative BSI while previous colonization with methicillin-resistant S. aureus BSI.” Previous colonization with what was associated with MRSA BSI?-

The phrase has been modified as follows: “Previous exposure to antibiotics, especially to carbapenems, was significantly associated with resistant Gram-negative BSI while previous colonization with methicillin-resistant S. aureus was associated with BSI due to this pathogen.”

Abstract: “Previous exposure to antibiotics or colonization can be cause of antibiotic resistant BSI.” This sentence does not make sense and needs to be rewritten for clarity. Do you mean “Previous exposure to anitibiotics or bacterial colonization can be the cause of antibiotic resistant BSI”? -

The phrase has been modified as follows: “Previous exposure to antibiotics or colonization by resistant pathogens can be the cause of antibiotic resistant BSI.”

Abstract: “This aspect stresses the need for careful local antibiotic stewardship and infec-tion control programs.” What aspect? -

The phrase has been modified as follows: “The significant evidence of center-level variations on AR, ICU admission and mortality, stress the need for careful local antibiotic stewardship and infection control programs.”

Introduction: “Historical studies showed that empirical antibacterial therapy…” In what context? In patients receiving chemotherapy or just SCT patients? It is not clear the way it is written in the introduction.

The phrase has been modified as follows: The introduction of empirical antibacterial therapy with the combination of anti-pseudomonal beta-lactam and an aminoglycoside in febrile neutropenic cancer patients has significantly decreased mortality

Introduction: “Following the results of a recent a meta-analysis 3monotherapy” The 3rd reference is not appropriately bracketed in the text. It should be denoted as [3].

The typo error has been corrected.

Introduction: “Antibiotic resistance (AR) is a worldwide threat, although vary considerably” It should say “varies” instead of “vary”

The sentence has been reformulated (see following q&a)

Introduction: “Antibiotic resistance (AR) is a worldwide threat, although vary considerably between institutions and geographical regions (https://atlas-surveillance.com/#/heatmap/resistance) and involves also . pediatric pa-tients receiving chemotherapy or allogeneic SCT, with consequent risk of inadequate initial empirical therapy of febrile neutropenia [6] causing increased morbidity and mortality [7-13]. “This sentence is difficult to follow and should be rewritten for clarity.

The phrase has been modified as follows: “Antibiotic resistance is a worldwide problem, although geographic and institution-level differences are observed (https://atlas-surveillance.com/#/heatmap/resistance). This phenomenon also affects pediatric patients receiving antineoplastic chemotherapy or allogeneic SCT [6] who become at risk of receiving and inadequate initial empirical therapy of febrile neutropenia, with an increased likelihood of complicated clinical course”

Introduction: “Epidemiological knowledges on antibiotic resistant bacteria and the consequences of invasive infections due to these pathogens in pediatric cancer and SCT patients are therefore mandatory in order to decide the best management strategy.” This sentence needs to be rewritten using proper grammar – epidemiological knowledge of antibiotic resistant bacteria…

The phrase has been modified as follows:

Knowledge of the epidemiology of antibiotic resistant bacterial infections and their consequences in pediatric cancer and SCT patients is therefore mandatory in order to identify the best management strategies.

Introduction: “Aims of the present study were to describe the proportion of antibiotic resistant non-common skin contaminants causing bloodstream infections (BSI) in pediatric patients re-ceiving antineoplastic treatments or SCT, to describe clinical risk factors associated with development of antibiotic resistance and to the risk of complicated clinical course, i.e. in-tensive care unit (ICU) admission and mortality.” Please change to “Aims of the present study were to describe the proportion of antibiotic resistant non-common skin contaminants causing bloodstream infections (BSI) in pediatric patients re-ceiving antineoplastic treatments or SCT, to describe clinical risk factors associated with development of antibiotic resistance and the risk of a complicated clinical course, i.e. in-tensive care unit (ICU) admission and mortality.”

The phrase has been substituted according to Reviewer’s suggestions

Materials and Methods: “For each BSI episode, demographic and disease-related variablese” There is a typo here with an “e” on the end of variables.

The typo error has been corrected.

Discussion: “Resistance among Gram-negatives was also about 25% for ciprofloxacin, overcom-ing the need for rethinking the practice of fluoroquinolone prophylaxis of febrile neutro-penia [21-24], which has indeed weakly recommended in the most recent pediatric guide-lines[25].” Please rewrite this sentence for clarity. Do you think the prophylaxis is the reason for resistance in this population? Should the prophylaxis be administered or not? It is unclear based on the way this sentence is written.

The phrase has been modified as follows: “The study also showed that about 25% of Gram-negatives were resistant to ciprofloxacin. This finding indicates the need for a rethink on fluoroquinolone prophylaxis of febrile neutropenia, which despite some effectiveness during chemotherapy courses (but not pre-engraftment neutropenia), has been associated with increased antibiotic resistance [21-24]. In this regard, it should be noted that in the most recent pediatric guidelines this procedure has received a weak recommendation about its use [25].

Discussion: “As a whole, all these observations underlying the need of effective, local antimicrobial stewardship and infection prevention and control programs [4,5,25-27] that could have an effective impact in the management of an infectious episode, including complicated clinical course.” This sentence has a lot of commas and is diddicult to understand. Please rewrite for clarity.

Taken together, all of these observations emphasize the need for the establishment at the local level of antimicrobial stewardship and infection prevention and control programs [4,5,25-27], which could also have a favorable impact on the management of infectious episodes and their complications

Reviewer 2 Report

The authors conducted important research analyzing data from several cancer centers in different countries. The article discusses the development of resistance to various antibiotics of bacterial or fungal infections and their relationship with a hold in intensive care units and mortality of pediatric cancer patients.

A conclusion is made about the probability of developing multidrug resistance to the most common infections, and the influence of the place of treatment (clinical center), and the combination of antibiotics for the successful therapy of pediatric patients after chemotherapy and/or hematopoietic stem cell transplantation is discussed.

The manuscript requires some editing. For example, a clearer description of the authors' conclusions. Below we provide some comments that, in our opinion, can improve this manuscript.

Major comments

Authors have to use the “Antibiotics” MDPI template for the text of the manuscript.

“Research ethics board approval was obtained at each site where it was required, according to local regulations”. ->  The Ethics Statement must be corrected to meet the requirements of the MDPI Antibiotics Journal.

There are no notes on the sources of funding for the work

Notes on Contribution of Authors is also missing

Minor comments

Abstract

“Bloodstream infections (BSI) are severe complication” -> Bloodstream infections (BSI) are severe complications.

“intensive care (ICU)” - the abbreviation is (IC) but is not (ICU). Do authors mean “intensive care unit”? The authors repeat the explanation of this abbreviation in the introduction. Hence, in the text of the abstract, they can be left their phrase intensive care without abbreviation.

abbreviation (IC) but not (ICU). Are the authors referring to the "intensive care unit"? The authors repeat the explanation of this abbreviation in the introduction. Therefore, in the text of the annotation, you can leave your phrase "intensive care" without abbreviation.

“stem cell transplant (SCT)” do authors mean “hematopoietic stem cell transplantation”? in the present, many types of stem cells are used for transplantation. The English medical literature also often uses the term hematopoietic transplant (HT) or (HSCT). https://pubmed.ncbi.nlm.nih.gov/32091673/

Introduction

“Epidemiological knowledges on antibiotic resistant bacteria” -> Epidemiological knowledge on antibiotic-resistant bacteria

Material and methods

 (https://nhsn.cdc.gov/nhsntraining/courses/2014/C18/page3299.html ) The reference is not working. Authors may have access to this resource by subscription, but not readers of the open-access journal.

Please edit the paragraph: “Antibiotic resistance (AR) is a worldwide threat, although vary considerably between institutions and geographical regions (https://atlas-surveillance.com/#/heatmap/resistance) and involves also . pediatric patients receiving chemotherapy or allogeneic SCT, with consequent risk of inadequate initial empirical therapy of febrile neutropenia [6] causing increased morbidity and mortality [7-13].”

Author Response

Author's Reply to the Review Report (Reviewer 2)

Thanks to the reviewer for the observation and the modifications suggested. Changes are written in yellow in the text

Abstract “Bloodstream infections (BSI) are severe complication” -> Bloodstream infections (BSI) are severe complications.

The typo error has been corrected

Abstract “intensive care (ICU)” - the abbreviation is (IC) but is not (ICU). Do authors mean “intensive care unit”? The authors repeat the explanation of this abbreviation in the introduction. Hence, in the text of the abstract, they can be left their phrase intensive care without abbreviation.

The abbreviation has been corrected,

Abstract abbreviation (IC) but not (ICU). Are the authors referring to the "intensive care unit"? The authors repeat the explanation of this abbreviation in the introduction. Therefore, in the text of the annotation, you can leave your phrase "intensive care" without abbreviation.

We left the abbreviation in the abstract since it was repeated some timed

Abstract “stem cell transplant (SCT)” do authors mean “hematopoietic stem cell transplantation”? in the present, many types of stem cells are used for transplantation. The English medical literature also often uses the term hematopoietic transplant (HT) or (HSCT). https://pubmed.ncbi.nlm.nih.gov/32091673/

The Reviewer is right, we apologize and corrected the abbreviation throughout the text

Introduction “Epidemiological knowledges on antibiotic resistant bacteria” -> Epidemiological knowledge on antibiotic-resistant bacteria

The phrase has been modified

Material and methods (https://nhsn.cdc.gov/nhsntraining/courses/2014/C18/page3299.html ) The reference is not working. Authors may have access to this resource by subscription, but not readers of the open-access journal. Please edit the paragraph: “Antibiotic resistance (AR) is a worldwide threat, although vary considerably between institutions and geographical regions (https://atlas-surveillance.com/#/heatmap/resistance) and involves also pediatric patients receiving chemotherapy or allogeneic SCT, with consequent risk of inadequate initial empirical therapy of febrile neutropenia [6] causing increased morbidity and mortality [7-13].”

We apologize for the error. the link has been corrected and verified (https://www.cdc.gov/nhsn/XLS/Common-Skin-Contaminant-List-June-2011.xlsx). Inserting it in google you get for free in automatic the excel sheet with the list of pathogens indicated as common skin contaminant

The paragraph has been modified (see also Reviewer #1)

Statements on Funding, Acknowledgements and Authors’ contributions have been added at the end of the main text, after references

Round 2

Reviewer 1 Report

The revised manuscript is greatly improved. All of my concerns have been addressed. I recommend that the manuscript be published.